



**Integrated ecohydrological hydrometric and stable water isotope data**
**of a drought-sensitive mixed land use lowland catchment**
**Doerthe Tetzlaff**[1,2]**, Aaron Smith**[1]**, Lukas Kleine**[1,2]**, David Dubbert** [1]**, Jonas Freymuller**[1]
**Hauke Daempfling**[1] **and Chris Soulsby**[3]
[1]IGB Leibniz Institute of Freshwater Ecology and Inland Fisheries Berlin, Berlin, Germany
[2]Humboldt University Berlin, Berlin, Germany
[3]Northern Rivers Institute, School of Geosciences, University of Aberdeen, UK
**Corresponding author:**
Doerthe Tetzlaff; doerthe.tetzlaff@igb-berlin.de

Abstract
Data from long-term experimental catchments are the foundation of hydrological sciences and
are crucial to benchmark process understanding, observe trends and natural cycles, and are
prerequisites for testing predictive models. Integrated data sets which capture all compartments
of our landscapes are particularly important in times of land use and climate change. Here, we
present ecohydrological data measured at multiple spatial scales which allows differentiation of
"blue" water fluxes (which maintain streamflow generation and groundwater recharge) and
"green" water fluxes (which sustain vegetation growth). There are two particular unique aspects
to this data set though: a) we measured water stable isotopes in the different landscape
compartments (that is in precipitation, surface water, soil, ground- and plant water); and b)
conducted this monitoring during the extreme drought of 2018 in Central Europe. Stable water
isotopes are so useful in hydrology as they provide "fingerprints" of the pathways water took
when moving through a catchment. Thus, isotopes allow to evaluate the dynamic relationships
between water storage changes and fluxes, which is fundamental to understanding how
catchments respond to hydroclimate perturbations or abrupt land use conversion. Second, as
we provide the data until 2020 one can also investigate recovery of water stores and fluxes after
extreme droughts. Last but not least: lowland headwaters are often understudied systems
despite them providing important ecosystem services such as groundwater and drinking water
provision and management for forestry and agriculture.



## 1. Introduction

Progress in scientific hydrology and provision of an evidence base for sustainable land and water management are only possible due to detailed, long-term observational data collected from long-term experimental watersheds (Hewlett et al., 1969; Robinson et al., 2013). Such experimental "outdoor laboratories" are invaluable scientific resources given the complexity of increasing pressures on water supplies (e.g. Cosgrove and Loucks, 2015), land use change (Neill et al., 2021) and the uncertain effects and non-stationarity of projected climate change (Milly et al., 2015).

Ecohydrology adopts an interdisciplinary approach to investigating interlinkages between the structure and function of ecological systems and the partitioning, flux and storage of fresh water (Guswa et al., 2020). Recent advances in monitoring and modeling have created manifold opportunities to address urgent ecohydrological questions on the importance of interlinkages of processes across the critical zone (CZ) - the dynamic, life-sustaining near-surface of the terrestrial earth that extends between the top of vegetation canopies, through the soil and into groundwater (Grant & Dietrich, 2017). Within the CZ concept, vegetation plays a central and dynamic role in partitioning incoming precipitation into "blue" water fluxes (streamflow generation and groundwater recharge) and "green" water fluxes which maintain vegetation growth (Evaristo et al., 2015).

To enhance ecohydrological process understanding in catchment systems, robust, multi-scale integrated data sets are required (Tetzlaff et al., 2021). In this regard, water stable isotopes and other tracers can help identify sources and pathways of water in the landscape and across the CZ to elucidate how different land use affects water partitioning between green and blue water fluxes (Dubbert and Werner, 2019; Tetzlaff et al., 2015). Importantly, water stable isotopes have enhanced the characterization of the celerity of hydrological fluxes in different CZ compartments, as well as quantifying the velocity of water particles and associated mixing relationships in the subsurface (Benettin et al., 2015; Birkel et al., 2011). Evaluating the dynamic relationships between water storage changes and fluxes is fundamental to understanding how catchments respond to hydroclimate perturbations, such as anomalous dry or wet periods, or abrupt land use conversion. This provides a more nuanced and integrated understanding of how key ecohydrological couplings may be at risk during long-term changes in blue and green water partitioning resulting from climate and land use change (Orth and Destouni, 2018). Such integrated understanding is important in the context of projected increases in air temperature, aridity, and in precipitation patterns, which may cause more variability in water availability threatening the sustainability of important ecosystem services (Okruszko et al., 2011). As an increase in drought frequency and severity is expected across Europe as the 21st century progresses, the development of effective and evidence-based amelioration measures to underpin sustainable and integrated land and water management policies for changing climatic conditions is urgently needed (Samaniego et al., 2018).



Consequently, integrated ecohydrological and stable isotope data sets targeted at understanding
the effects of different types of environmental change have outstanding potential, not least
because interdisciplinary environmental research tends to give unanticipated insights (Burt,
1994). Such integrated data streams allow identification and quantification of the linkages
between rainfall, soil moisture, groundwater and runoff generation, facilitating deeper
understanding of flood and drought risk in different types of landscapes and under different land
use management (Huntingford et al., 2014).
Water resources in the extensive, glacially formed, lowland landscape of northern Europe,
including the North German Plain (NGP) sustain food production (Gutzler et al., 2015; Barkmann
et al., 2017) and water supplies to large cities like Berlin. Interestingly, such lowlands catchments
are still relatively understudied compared to more upland headwater landscapes with stronger
topographic controls on drainage of surface and subsurface water (Devito et al., 2005). In low
elevation catchments across the NGP, streams are usually groundwater-dominated, but the
temporal and spatial heterogeneities in the hydrological functioning of these catchments are still
not fully understood (Boulton and Hancock, 2006). For example, there is still a limited evidence
base for quantifying how drought affects groundwater recharge and stream flow generation in
lowland areas in Central Europe, including the cessation of flow during the summer (Germer et
al., 2011).
To help address these knowledge gaps, here, we present a comprehensive set of ecohydrological
hydrometric and stable water isotope data of two years of data for the Demnitzer Mill Creek
catchment, NE Germany. The data set is unique in its integrative characteristics; that the different
compartments of the CZ were sampled across a mesoscale catchment in terms of their isotopic
signature and supporting ecohydrological data. By coincidence, these first two years, of what will
be a long-term study, captured the changing impacts of a prolonged drought period (2018-2020)
with a strong negative rainfall anomaly that became the most severe regional drought so far in
the 21$^{st}$ century (Kleine et al., 2021a). The data allow the effects of droughts (and their
persistence) on water storage, fluxes and age dynamics in the CZ to be investigated (Smith et al.,
2022). Our objective here is to provide this high spatio-temporal resolution ecohydrological
dataset to improve understanding of the storages and flow pathways of both blue and green water
across processes at the larger catchment scale in lowland catchments. We are continuing these
observations to assess long-term climatic trends at the drought sensitive region of NE Germany.

**2. Site description**

Figure 1: The Demnitzer Mill Creek catchment and its location within Europe and Germany. Hexagonal points (⬣) are measurement locations in the catchment and the star (★) are meteorological measurements by the German Weather service (DWD Deutscher Wetterdienst).

**Figure 1: The Demnitzer Mill Creek catchment and its location within Europe and**
**Germany. Hexagonal points (⬣) are measurement locations in the catchment and the star**
**(★) are meteorological measurements by the German Weather service (DWD Deutscher**
**Wetterdienst).**




**Table 1** - Overview of the properties of the Demnitzer Millcreek catchment at the catchment
outlet. Overview includes physiological characteristics, landuse, and geology.

| Area (km2) | 66.39 | Topographic Relief (m) | 50.23 |
|---|---|---|---|
| Runoff Ratio | 0.10 | Mean Slope (%) | 1.98 |
| | | | |
| Landuse (%) | | Geology (%) | |
| Mixed Forest | 1.0 | Base moraine | 35.5 |
| Conifer Forest | 29.2 | End moraine | 2.3 |
| Broadleaf Forest | 6.0 | Deposits of glacial valleys | 6.9 |
| Peat | 0.7 | Peat Fen | 5.9 |
| Pasture | 10.2 | Periglacial/fluvial deposits | 16.3 |
| Agricultural/arable land | 50.4 | Glacial/fluvial deposits | 31.1 |
| Urban | 2.5 | Sandy peat fen | 2.0 |

The data presented here were monitored in the Demnitzer Millcreek catchment (DMC) located in
NE Germany (52°23′N, 14°15′E; Figure 1). The DMC is a lowland drought-sensitive area south
east of Berlin, the German capital, and situated in the NGP. The region has high socio-economic
significance through the provision of numerous ecosystem services; including food security,
timber production, groundwater recharge and river flow generation which sustains drinking water
supplies for Berlin (Kleine et al., 2021a). The original motivation behind establishing DMC as an
observatory in 1990 was to investigate the impact of agricultural pollutants on surface water
quality (Gelbrecht et al., 2000, 2005).
The hydroclimate is temperate with warm, humid summers (Kottek et al., 2006). Mean annual
precipitation and air temperature are 567 mm yr$^{-1}$ and 9.6°C, respectively (DWD, 2020, for 2006-
2015). Seasonal contrasts are characterized by higher summer precipitation, mainly from high
intensity, convective events; and slightly lower precipitation during frequent, frontal rainfall events
in winter. The landscape was shaped by the last glaciation (Weichselian); soils are predominantly
sandy and formed on glacial and fluvial deposits (Kleine et al., 2021b). The catchment is
dominated by groundwater and likely had little surface runoff before human intervention.
Previously, numerous peat fens and freshwater lakes in hollows existed, but these were drained
during a long historic evolution of anthropogenic management (Nützmann et al., 2011). Land use
is currently dominated by farming and forestry (Kleine et al., 2020; Smith et al., 2020c). The
catchment is also relatively sparsely populated, and has recently experienced recolonization of
beaver (Smith et al., 2020a), wolf (Vogel, 2014) and even sporadic sighting of elk (Martin, 2014).
Maintenance of crucial ecosystem services in the landscape is dependent on sufficient seasonal
precipitation input to sustain adequate soil moisture levels in the rooting zone to support crop and
tree growth (Drastig et al., 2011); and acceptable groundwater recharge to sustain groundwater-
surface water exchanges. However, high (~90 %) proportions of evapotranspiration, particularly
from forested areas and poor water retention in the widespread sandy soils (Smith et al., 2021),
result in catchment drought sensitivity (Kleine et al., 2020). Further, increased flow disconnections
and fragmentation of the stream network occurs during droughts (Kleine et al., 2021a; Smith et
al., 2021).



**3. Data and instrumentation overview**

**3.1 Instrumentation overview**

A fully automatic weather station (AWS) was installed and has been operated in Hasenfelde (Hf, Figure 1) since April 2018, including radiation, air temperature, relative humidity, precipitation and ground heat flux every 15-minutes. A modified autosampler (ISCO 3700, Teledyne Isco, Lincoln, USA) was installed nearby to collect daily samples of precipitation to supplement the AWS. Weekly cumulative precipitation was additionally collected at four locations nested from north to south in the catchment: Marxdorfer St., Demnitz Mill, Bruchmill, and Berkenbruck (Figure 1&2) from July 2018 to April 2020. Measurements of throughfall were collected under the canopy at Forest A at five locations (Forest A1-5) within a 10m square fenced area. Throughfall was collected using simple rain gauges (Rain gauge kit, S. Brannan & Sons, Cleator Moor, UK; https://doi.org/10.18728/igb-fred-623.0)

Soil moisture and temperature profiles were established at Forest A (FA) and Grass A (GA) in June 2018 with 18-sensors per site (SMT-100, Umwelt-Geräte-Technik GmbH, Müncheberg, Germany). The sensors were distributed equally at soil depths of 20, 60, and 100cm at each site (i.e. three sensors per depth), measuring every 15-minutes (https://doi.org/10.18728/igb-fred-623.0).

Sap flow measurements were established in 12 trees at Forest A including Scots Pine (*Pinus sylvestris*), European Oak (*Quercus robur*), common hazel (*Corylus avellana*), and Red Oak (*Quercus rubra*). Measurements were conducted using 2-4 radially installed thermal dissipation-based sap flow sensors (TDP probes, Dynamix Inc., Houston, TX, USA). Sap flow measurements were recorded every 15 minutes (https://doi.org/10.18728/igb-fred-623.0).

Stream water level was established at four locations within the catchment; Peat North, Bruchmill, Demnitz Mill, and Berkenbruck (https://doi.org/10.18728/igb-fred-623.0).The water level was established by IGB Leibniz Institute of Freshwater Ecology and Inland Fisheries and recorded with divers (Micro 10m and Baro) at Peat North and Demnitz Mill, and at Bruchmill (Van Essen Instruments). The divers utilized at each site include an internal atmospheric pressure correction (AquiLite ATP 10, AquiTronic Umweltmeßtechnik GmbH, Kirchheim/Teck, Germany). Water level measurements began at Demnitz Mill in 1986, and in January and June 2018 for Peat North and Bruchmill, respectively. Water level has been recorded since 1982 at Berkenbruck using pressure transducers and was established and collected by the Landesamt fur Umwelt. Channel stability at Demnitz Mill and Berkenbruck has permitted rating curve development to translate water level measurements to discharge. Stream water level at Bruchmill was supplemented with daily stream water samples for stable water isotope analysis collected from an autosampler (ISCO 3700, Teledyne Isco, Lincoln, USA). The autosampler was established in December 2018 (https://doi.org/10.18728/igb-fred-623.0).

Groundwater level divers were installed at five locations throughout the catchment in 2001 (GW3, GW4, GW5, GW7, and GW8) (Figure 1&2). Groundwater level at each site was measured every four hours with an AquiLite ATP-10 diver (AquiTronic Umweltmeßtechnik GmbH, Kirchheim/Teck,



Germany) with internal correction for atmospheric pressure (https://doi.org/10.18728/igb-fred-
187 623.0)


### 3.2 Isotope sampling overview

Manual sampling from different locations and different water cycle / landscape compartments
supplemented the autosamplers installed for precipitation at Hasenfelde and for stream water at
Bruchmill. Samples were taken from the weekly cumulative precipitation and throughfall (Forest
A) for each location (Figure 2). Further, monthly samples of soil water were taken at 6 depths
(2.5, 7.5, 15, 30, 60, 90 cm) in triplicate for Forest A and Grass A. This was complemented by
synoptic, spatially distributed sampling of the upper 30cm in 2019. Samples were placed in a
sterile zip-lock bag (CB400-420siZ, Weber Packaging GmbH, Güglingen, Germany) and
analyzed using the direct water vapour equilibrium method (Wassenaar et al., 2008). Weekly grab
samples of stream water were taken at all nested stream water locations (eight locations).
Groundwater isotopes were sampled at six groundwater wells, including two with continuous
groundwater level measurement (GW3, GW8). Groundwater levels at the other sites (GW DA,
GW6, GW WLV, GW BB) were periodically recorded. Vegetation isotopic sampling was
conducted by taking twig samples from different vegetation in Forest A and samples of the non-
green stem of the grass at site Grass A. Vegetation samples were stored at -20°C after sampling
until analysis. Reference for all isotope samples is https://doi.org/10.18728/igb-fred-623.0.
A layer of paraffin was added to the bottom of all autosampler containers to prevent evaporation
and fractionation from collected water. Autosamplers are emptied each week. Collected weekly
precipitation, throughfall, stream water, and groundwater were sealed and refrigerated until
isotopic analysis.
All liquid water samples ($P_{iso}$, $THR_{iso}$, $Q_{iso}$, $GW_{iso}$) were filtered (0.2 μm, cellulose acetate, Lab
Logistics Group GmbH, Meckenheim, Germany) and cooled before beeing analyzed using Cavity
Ring-Down Spectroscopy (CRDS, L2130-i, Picarro, Inc., CA, USA). Additionally, the CDRS was
used for the analysis of the to direct liquid-water equilibrium method for soil water. Vegetation
samples were extracted in January 2020 using the cryogenic extraction method given in Dubbert
et al. (2013, 2014) and analyzed with the CDRS.





**Table 2 – Site locations in DMC, including site name, coordinates, data collected, start**
**and end dates, and resolution. N/A indicates not applicable, P is precipitation, GW is**
**groundwater level, THR is throughfall, Ts is soil temperature, va is wind speed/direction,**
**Ta is air temperature, Pa is air pressure, RH is relative humidity, NR is net radiation, Sap**
**is sap flow, and subscript iso indicates isotopic sampling. AWS indicates measurements**
**of P, va, Ta, Pa, RH, and NR**

| Site Name | ID | Location (UTM 33N) | | Data Type | Installation/Start Date | Discontinued/End Date | Resolution | |
|---|---|---|---|---|---|---|---|---|
| | | Latitude | Longitude | | | | Temporal | Spatial |
| Marxdorfer St. | Marxdorfer St. | 5810076 | 449773 | P, $P_{iso}$, $Q_{iso}$, $T_s$ | Jan 10, 2018 ($Q_{iso}$) Jul 9, 2018 (P&$P_{iso}$) Aug 16, 2019 ($T_s$) | Jun 2, 2020 (P&$P_{iso}$) Jul 11, 2020 ($T_s$) | Weekly (P, $P_{iso}$ & $Q_{iso}$) 15-min ($T_s$) | $T_s$ (5cm) |
| Hasenfelde | Hf | 5809705 | 446068 | P, $P_{iso}$, $v_a$, $T_a$, $P_a$, RH, NR, $T_s$ | Mar 17, 2018 (AWS) Jul 12, 2018 ($P_{iso}$) Aug 16, 2019 ($T_s$) | Jul 11, 2020 ($T_s$) | 15-min (AWS & $T_s$) Daily ($P_{iso}$) | AWS (2m) $T_s$ (5cm) |
| Groundwater DA | GW DA | 5808335 | 447527 | $GW_{iso}$ | Apr 16, 2019 | N/A | Monthly | N/A |
| Peat North | PN | 5807703 | 447474 | $Q_{iso}$ | Jan 10, 2018 | N/A | Weekly | N/A |
| Groundwater 3 | GW3 | 5807499 | 447582 | GW | Jan 10, 2001 | N/A | 4-hour | N/A |
| Groundwater Ringwall | GW4 | 5807247 | 447233 | GW, $GW_{iso}$ | Feb 22, 2001 (GW) Sep 11, 2018 ($GW_{iso}$) | N/A | 4-hour (GW) Monthly ($GW_{iso}$) | N/A |
| Groundwater 5 | GW5 | 5807099 | 447490 | GW | Jan 10, 2001 | N/A | 4-hour | N/A |
| Peat Ditch | Peat Ditch | 5806364 | 446487 | $Q_{iso}$ | Mar 21, 2018 | N/A | Weekly ($Q_{iso}$) | N/A |
| Groundwater Peat Ditch | GW8 | 5806320 | 446488 | GW, $GW_{iso}$ | Jan 10, 2001 (GW) Aug 15, 2018 ($GW_{iso}$) | N/A | 4-hour (GW) Monthly ($GW_{iso}$) | N/A |
| Groundwater 7 | GW7 | 5806307 | 447726 | GW | Feb 22, 2001 (GW) | N/A | 4-hour (GW) | N/A |
| Groundwater 6 | GW6 | 5806274 | 447678 | $GW_{iso}$ | Sep 11, 2018 | N/A | Monthly | N/A |
| Peat South | Peat South | 5806262 | 447712 | $Q_{iso}$, $T_s$ | Jan 10, 2018 ($Q_{iso}$) Aug 16, 2019 ($T_s$) | Jul 11, 2020 ($T_s$) | Weekly ($Q_{iso}$) 15-min ($T_s$) | $T_s$ (5cm) |
| Forest A | FA | 5805520 | 445731 | Sap, SM, $SM_{iso}$, THR, $THR_{iso}$, $T_s$ | Apr 21, 2018 (Sap) Jun 15, 2018 (SM & $T_s$) Oct 18, 2018 ($SM_{iso}$) Jul 11, 2018 (THR & $THR_{iso}$) | Nov 1, 2018 (Sap) N/A (SM) Jul 16, 2019 ($SM_{iso}$) May 19, 2020 (THR & $THR_{iso}$) | 15-min (Sap) 15-min (SM & $T_s$) Monthly ($SM_{iso}$) Weekly (THR & $THR_{iso}$) | 12 Trees (Sap) SM & $T_s$ (6 sites, 20, 60, 100cm depths) |



| | | | | | | | THR & THR$_{iso}$ (5 sites) |
|---|---|---|---|---|---|---|---|
| Grass A | GA | 5805125 | 445495 | SM, SM$_{iso}$, T$_s$ | Jun 15, 2018 (SM & T$_s$) Oct 18, 2018 (SM$_{iso}$) | Jul 16, 2019 (SM$_{iso}$) Jan 7, 2020 (SM & T$_s$) | 15-min (SM & T$_s$) Monthly (SM$_{iso}$) | SM & T$_s$ (6 sites, 20, 60, 100cm depths) |
| Bruchm ill | Bruchm ill | 5805088 | 445459 | P, P$_{iso}$, Q$_{iso}$ | Jan 10, 2018 (Q$_{iso}$- weekly) Dec 28, 2018 (Q$_{iso}$- daily) Jul 9, 2018 (P&P$_{iso}$) | Dec 28, 2018 (Q$_{iso}$- weekly) Jun 2, 2020 (P&P$_{iso}$) | Weekly (P & P$_{iso}$) Daily (Q$_{iso}$) | N/A |
| Ground water WLV | GW WLV | 5803322 | 445982 | GW$_{iso}$ | Sep 20, 2018 | N/A | Monthly | N/A |
| Demnit z Mill | Demnit z Mill | 5802298 | 445188 | P, P$_{iso}$, Q, Q$_{iso}$ | Jan 10, 2018 (Q$_{iso}$) Jul 9, 2018 (P&P$_{iso}$) Feb 22, 2011 (Q) | Jun 2, 2020 (P&P$_{iso}$) | Weekly (P, P$_{iso}$ & Q$_{iso}$) 4-hour (Q) | N/A |
| Fox Bridge | Fox Bridge | 5801469 | 444189 | Q$_{iso}$ | Jan 10, 2018 | N/A | Weekly | N/A |
| Ground water Berken bruck | GW BB | 5799862 | 444611 | GW$_{iso}$ | Jan 21, 2019 | N/A | Monthly | N/A |
| Berken bruck | Berken bruck | 5799604 | 444737 | P, P$_{iso}$, Q, Q$_{iso}$, T$_s$ | Nov 1, 1982 (Q) Jan 10, 2018 (Q$_{iso}$) Jul 9, 2018 (P&P$_{iso}$) Aug 16, 2019 (T$_s$) | Jun 2, 2020 (P&P$_{iso}$) Jul 11, 2020 (T$_s$) | Daily (Q) Weekly (P, P$_{iso}$ & Q$_{iso}$) 15-min (T$_s$) | T$_s$ (5cm) |






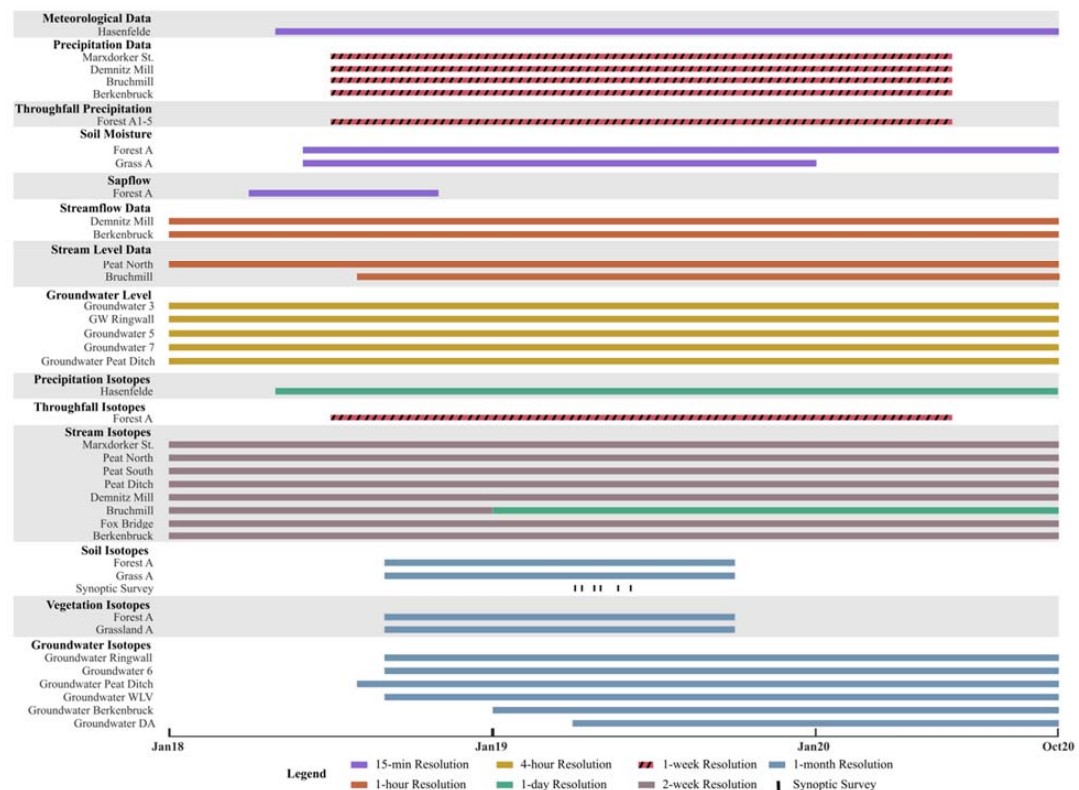

**Figure 2: Spatial data availability and temporal resolution (colour code) of the measurements within the Demnitzer Millcreek Catchment including, meteorological, soil, vegetation, stream, and groundwater hydrological and isotope data sets.**



### 4. Precipitation and throughfall amount

Monitoring for precipitation commenced in the 2018 summer drought when low rainfall inputs continued through the following winter (Figure 3a). Large rainfall events (>20 mm/d) were relatively rare and mostly summer convectional storms. Even by summer 2020, most months had below average rainfall. Throughfall at the Forest A site typically was 70-90 % of incident rainfall, with higher interception losses in low intensity summer storms and lowest in winter or high intensity summer storms. Heterogeneity in throughfall was marked (Figure 3b), emphasizing the importance of the forest canopy is redistributing net rainfall to the forest floor.

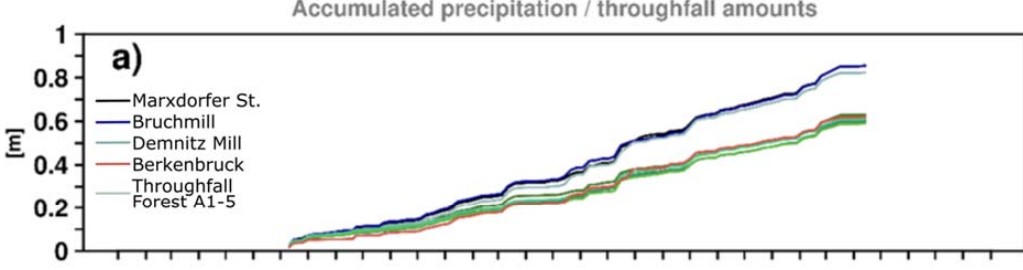

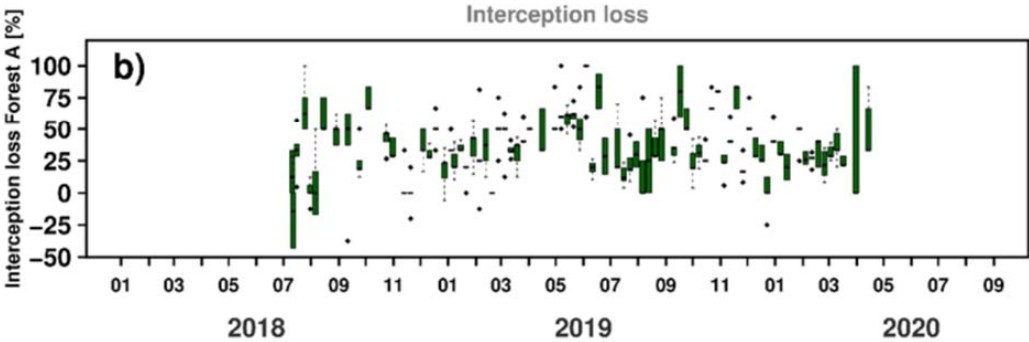

**Figure 3: (a) Cumulative precipitation and throughfall at multiple locations throughout the catchment. Throughfall was collected weekly at Forest A with (b) five samplers (1-5) distributed throughout the 10m square fenced region.**



**5. Catchment response data**

Rainfall fluxes mostly drove short term soil moisture variations (Figure 4a, c); which were more responsive in the upper soil layers (at 20 cm) than deeper layers. Variability was also more sensitive under forested land cover, where soils are sandier, more structured and effective rainfall is lower due to interception losses. Seasonality in evapotranspiration (usefully indexed by sapflow in Figure 4b) modulated the effects of rainfall on soil moisture storage. Seasonal soil moisture dynamics also governed groundwater recharge and variation in groundwater levels, which had an annual range of ~1.5 m at well G3 and ~1m at the peat ditch well (Figure 4d). Despite clear winter recharge and spring drawdown in each well, peak winter and summer levels were lower in 2019 and 2020 compared to 2018 indicating the cumulative "memory effects" of the drought. This was also evident in the stream hydrograph with very low discharge peaks in 2019 and 2020, which also had prolonged periods where flow ceased in the summer, particularly at Berkenbruck. Thus, despite winter soil moisture replenishment, this was insufficient to match long-term groundwater recharge. These different correlations underline the added value of simultaneous data from long-term study sites on transpiration, soil water, groundwater and stream flow as droughts develop (Smith et al., 2022).

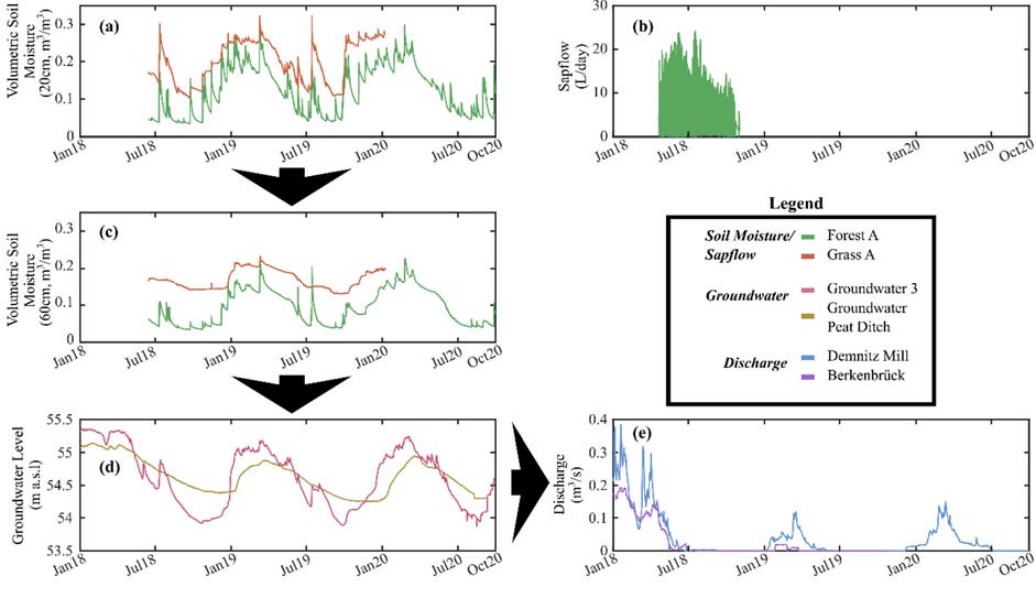

**Figure 4: (a) Shallow and (c) deep soil moisture, (b) sapflow, (d) groundwater levels and (e) discharge within the Demnitzer Millcreek catchment. Arrows show connections between layers and fluxes. *Groundwater 3 is within the wetland and Groundwater Peat Ditch is outside the wetland (near Forest A and Grass A, Fig. 1).**



**6. Stable water isotopes**
Stable water isotope signatures in precipitation showed high day-to-day variability superimposed
on strong seasonality; with more depleted values in winter and more enriched values in summer
(Fig 5a). Interestingly, weekly throughfall signatures were very similar to the (weekly and daily)
precipitation signal showing no strong signs of evaporative fractionation during canopy storage
(Fig 5b). This likely reflects the high intensity nature of most summer rainfall, which affords limited
opportunity for canopy evaporation. Streamwater signatures at all nested sites showed similar
seasonality but much more damping in the signal (Fig. 5c). Groundwater was most damped, and
similar in composition to streamflow during winter (Fig 5d). In summer, sites downstream of
Marxdorfer Strasse showed evidence of evaporative fractionation from either the channel network
or riparian soils and plotted below the meteoric water line before stream flow ceased. Monthly soil
water samples showed higher variability in isotopic composition under forest than under grass,
mainly reflecting soil characteristics with more retentive, loamy and wetter soils at the grassland
site buffering the effects of rainfall inputs. At both sites, seasonal variation in isotopic composition
tracked precipitation, though in deeper soil water samples were more damped. Vegetation
samples from the oaks showed higher variation than from grass.

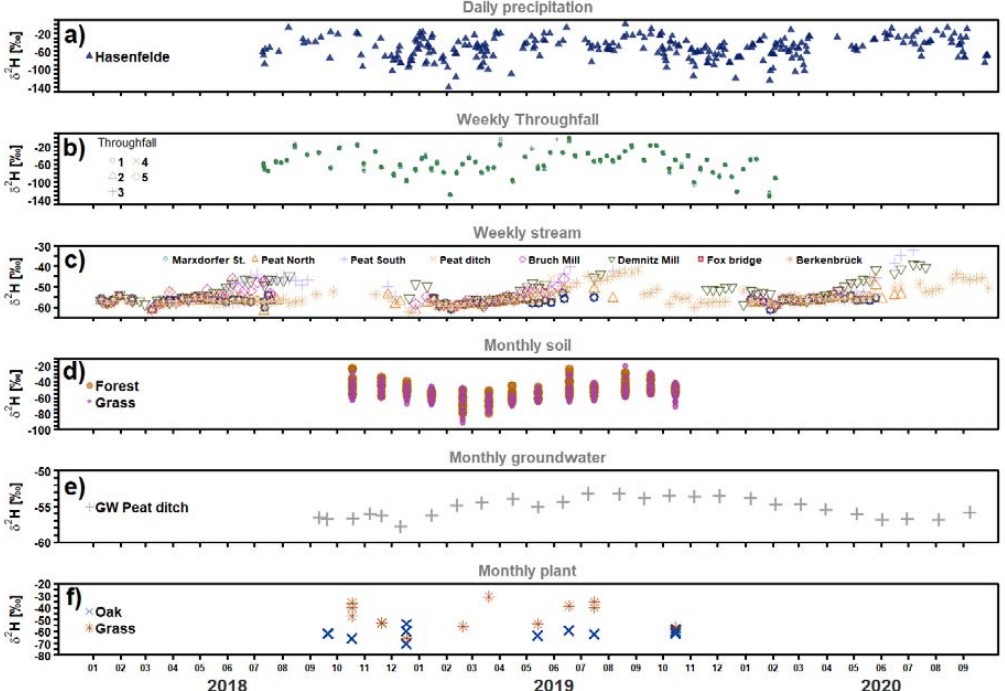

**Figure 5: Time series of deuterium (δ²H) in (a) precipitation, (b) throughfall (Forest A), (c)**
**stream water, (d) soil water, (e) groundwater and (f) plant samples at various locations in**
**the catchment.**




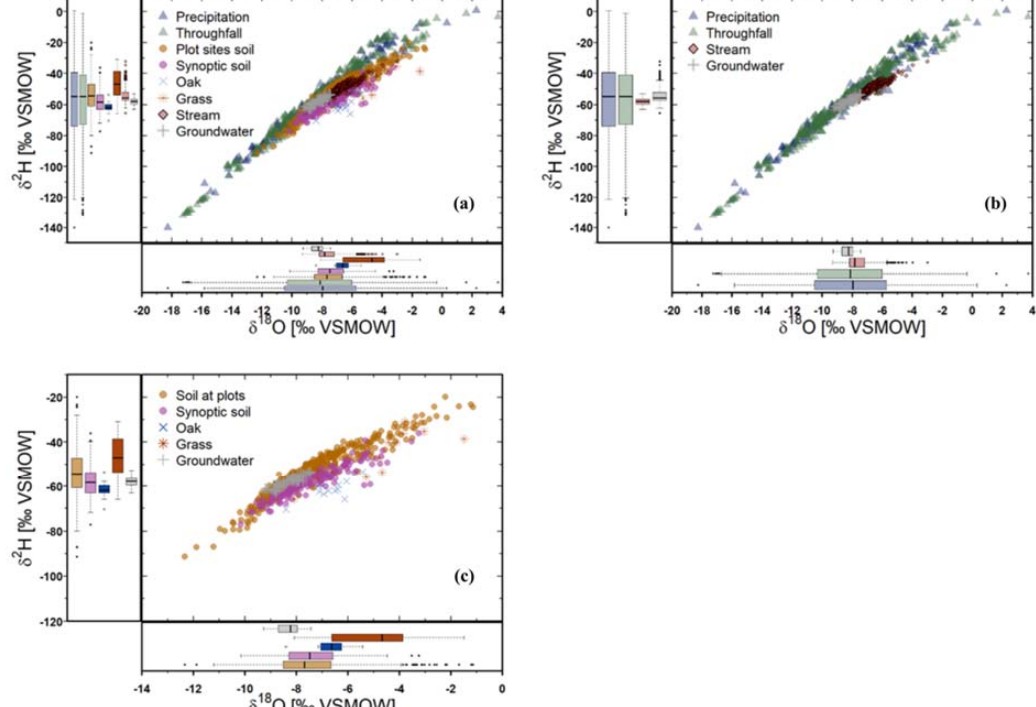

**Figure 6: Dual isotope space ($\delta^2$H-$\delta^{18}$O) plots for (a) all measured isotopic datasets, (b)**
**precipitation, throughfall, stream, and groundwater, and (c) soil (multiple depths),**
**synoptic soil survey (upper 30cm), vegetation, and groundwater.**

Differences in the isotope dynamics of different critical zone compartments are shown in dual
isotope space in Figure 6a. The damping of precipitation in groundwater and streamflow is
apparent, as is the fractionation of more enriched summer stream flow samples (Figure 6b). The
role of the soil in partitioning water is apparent from the overlap between deeper soil horizons
and groundwater which were both more weighted to winter precipitation – when recharge is
greatest (Fig 6c). Xylem water in oaks and grass tended to show the effects of fractionation,
which was most marked in the oaks and may point to different soil water sources of root uptake.



## 7. Data availability

All data presented in this paper are available from the IGB open data repository FRED https://fred.igb-berlin.de/data/package/622 (Tetzlaff et al., 2022). The data is published with detailed metadata (https://doi.org/10.18728/igb-fred-623.0) and contact information for any further questions. There is a readme section per each dataset. We also included a digital elevation model, shapefile of the catchment boundary and the station locations.

## 8. Summary

The integrated data set presented in this paper is unique because: (1) it captures complicated ecohydrological dynamics over two years during an exceptional drought (in 2018/2019) in Central Europe; (2) the different compartments of the critical zone were monitored through stable water isotope data and complimentary ecohydrological data for contrasting land use and (3) multi-scale, nested catchment time series were derived. In total data from 49 time series / data sets are available. The data are quality controlled. We included meteorological data and precipitation and throughfall amount. Catchment response data include stream discharge at the catchment outlet and another nested site, and stream level data at two further sites; soil moisture from multiple depths at two locations (two different landuses), groundwater level data at five locations and sapflow measurements from one forest location. Stable water isotope data include precipitation water, throughfall, streamwater at eight sites, soil water isotopes from two sites plus spatially distributed samples of upper soils, vegetation samples at two locations and groundwater at six locations. Data continue to be collected and updated data sets will be published based on available resources.

As such, these data provide an excellent, integrated ecohydrological perspective on the drought response of a lowland agricultural landscape. Such data are of course important in their own right, but are equally invaluable for challenging environmental models as constraints on internal model function that can be used to increase confidence in the use of models in projecting the impacts of future change. Integrated data like the ones summarised here are also important for a range of scientific questions that are growing in importance as the effects of climate change become more apparent. These include understanding how do droughts develop and propagate through components of hydrological systems and compartments of the critical zone? What are the effects of land cover on this propagation and how does it affect water cycling in vegetation? How long does recovery of different system components take once rainfall anomalies become positive? How resilient are different critical zone compartments or entire landscapes against climate extremes such as droughts? Hopefully, this data set will be used by scientists to increase understanding on critical issues such as what are the water footprints of alternative land uses and how can these be reduced whilst maintaining societal needs. This will help to contribute to the development of more sustainable and resilient land and water management policies that will be needed in the face of increased longevity and frequency of droughts.



**Author contributions**: AS and LK prepared the data sets. Datasets were collected by LK and
JF. Isotope data were analysed by DD. DT, CS, AS prepared the manuscript with contributions
from all co-authors.
**Competing interests**: The authors declare that they have no conflict of interest.
**Disclaimer**: any reference to specific equipment types or manufacturers is for informational
purposes and does not represent product endorsement. IGB is an equal opportunity provider.

**Acknowledgements**:
We acknowledge the BMBF (funding code 033W034A) which supported the stable isotope 650
laboratory at IGB. Funding for DT was also received through the Einstein 652 Research Unit
"Climate and Water under Change" from the Einstein Foundation Berlin and 653 Berlin
University Alliance.

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
