# Peer review of "Integrated ecohydrological hydrometric and stable water isotope data"

_Earth System Science Data, 2022_

## Author Comment (AC1)

Reviewer 1
General Comments:
This rich hydrologic dataset will be useful for future analyses of long-term changes in hydrologic functioning in the DMC catchment. The fact that it captured a significant drought period makes it especially unique. The data could also be used for inter-site comparisons to better understand how landscape characteristics influence hydrologic behaviour in response to disturbance (e.g., climatic extremes).

The hydrologic data are useable in the current format and size, although would require significant data wrangling to prepare for specific analyses. The GIS data are useable in the current format and size. I provide some specific comments below to improve the accessibility and useability of the hydrologic data.

The article is well structured and clear. The data collection and analyses methods are described in sufficient detail for the most part. All references to other data sets or papers seem appropriate. I have provided minor technical comments below that could help to clarify certain aspects of the data collection and analyses methods. The figures and tables are high quality and I only have a few minor suggestions for improving them.

Despite my suggestions for improvement, by reading the article and downloading the dataset I am confident that I could use the data.
** thank you for this positive assessment of our data set, its uniqueness and also the general manuscript. Please find our detailed responses below.

Specific Comments:
1. The majority of the digital object identifiers provided in the manuscript take the reader to the Demnitz_Isotope_Hydrological_Data page (number 623) on the IGP Freshwater Research and Environmental Database website. There a description of the available data is given; however, it is not possible to download the data here. In the 'Data files' section, it says that a user needs to make a request to the three contacts to access the file. However, in section 7 of the manuscript, the reader is directed to data package 622, where the data can be downloaded. I recommend that the 'Data files' – Actions column on package 623 be updated with a link to 622. Or to make the process even more streamlined, just eliminate 623 altogether. It's not clear why there are two data packages.
** we would like to clarify: the current data set that is applicable to this paper is FRED Package 622 (https://fred.igb-berlin.de/data/package/622). FRED Package 623, which corresponds to the DOI 10.18728/igb-fred-623.0, was an earlier version of this dataset. Of course, a new DOI will be assigned before publication of this paper; however, it cannot be assigned yet because there may still be improvements made to the dataset, while assigning a DOI will "freeze" the dataset. We apologize for the confusion and can assure you that it will be alleviated by the assignment of a new DOI and its use in the final manuscript once the dataset is finalized.

2. There are no apparent inconsistencies in the data and no unusual data formats, however, I think there could be some improvements in the way the data is organized that could reduce the pre-processing time for a user. My recommendations for improving the data organization are to:
   a. Make sure that all columns have a header (e.g., on a few sheets the year, date, or date_time header is missing, colP header on 'Forest_A_Isotopes' is missing, etc.).
   b. Make sure to standardize the column headers across sheets and I recommend using lower case with an underscore separator between words to keep the headers simple (e.g., don't use different types of brackets around the units).

     c.  Consider integrating different levels of column headers (e.g., instead of 'Average Moisture [m3/m3]' and '20cm', use 'avgsm_m3_m3_20cm'… or something like that). On the readme page the authors could create a data dictionary that translates these abbreviated headers for the user.

     d.  Move all the 'readme' textboxes off of the data sheets and onto a single readme sheet (use the 'General Readme' sheet).

     e.  Use ISO 8601 date format consistently on all sheets.

     f.  Make sure that blank cells in data columns are replaced with 'NaN'.

** we have adapted all these suggestions and improvements. We will upload a new data file in the revision. We will also consider the comment made below (on l.312).

Technical Comments:

38: Remove one of the 'long-term'.
** we will correct this in the revision

43: Consider '… to investigate links between…'.
** we will correct this in the revision

46: Consider '… importance of links between…'.
** we will correct this in the revision

50-1: I think the quotation marks should go around "blue water" and "green water", not just around the colours.
** we will correct this in the revision

53: What do the authors mean by robust data sets? I am familiar with the concept of robust statistics, but unsure what constitutes a robust data set.
** we mean "quality controlled" monitoring data. We would prefer to leave this text as it is.

81: NGP only used three times – consider just spelling it out.
** we will change this in the revision

93: Spell out northeast.
** we will correct this in the revision

93: Consider removing 'its integrative characteristics;'.
** We would prefer to leave this text as it is, as they are integrative data (integrating information content of very different environmental processes)

103: I recommend explaining here what the characteristics of a drought sensitive region are. The authors could also consider mentioning whether or not the data can potentially be used to understand the hydrologic functioning of other drought sensitive regions beyond northeast Germany.
** yes that's a good suggestion. We will explain this in the revision.

Fig. 1: Instead of one symbol for all measurements, I recommend using different symbols to represent meteorological, soil, sap flow, groundwater, and isotope measurements.
** we will correct this. Please find the revised / new figure 1 already copied below.

134: Consider replacing '… historic evolution…' with '… history…'.

** yes good point, we will correct this in the revision.

141: '… high (~90 %) proportions of evapotranspiration…' is a bit confusing – consider '… high water losses due to evapotranspiration (~ 90 % of total precipitation)…'.
** yes much better: we will correct this in the revision.

141-3: Here, the authors explain some of the characteristics of a drought sensitive region. I recommend these be mentioned earlier when the concept is first introduced.
** yes we will do this as suggested.

150: Was it net radiation that was measured?
** yes, we will correct this.

157: Were standard rain gauges used? If so, replace 'simple' with 'standard'.
** yes, we will correct this.

179-80: Consider reorganizing the sentence, e.g., Water samples were also collected at Bruchmill using an autosampler (ISCO…) and analyzed for stable water isotopes.
** we will reformulate this sentence.

197-8: Are these nested sites marked on the map? If so, it needs to be clear which sites they are.
** we will correct Fig 1 and make this clearer. We will also change the symbols for the different sites. We will also add a reference to fig 1 here.

200-1: I recommend putting this information on groundwater levels in the previous section.
** yes this was misleading. Thanks for pointing this out. We will remove the info on groundwater level measurements here (as its already stated in the previous section).

202: Were any twigs sampled or did they have to have certain characteristics?
** we are not sure what the reviewer means here as we do state ".. by taking twig samples from different vegetation in Forest A and samples of the non-green stem of the grass.."?

207-8: What was the average and max storage time?
** usually the samples are analysed within one week in our isotope lab. We will add this information.

209: Either introduce these abbreviations when the water source is first mentioned in the isotope section or add the water source in front of each abbreviation in the brackets.
** good point. We will correct this (and add water source in front of each abbreviation in the brackets).

211: Replace 'CDRS' with 'CRDS'.
** we will correct this. Well spotted.

212: Consider revising '… to direct liquid water equilibrium method for soil water' to '… soil water extracted via the direct liquid water equilibrium method'.
** we will change this.

Table 2: Instead of repeating the abbreviations in brackets after dates, consider creating sub-rows from the data type column to the far-right column of the table, i.e., a sub-row for each data type.
** good idea. We will implement this.

Table 2: I think for some of the spatial resolution entries the direction (vertical or horizontal) should be given.
** We will implement this.

Figure 2: In the caption consider replacing 'Spatial data availability…' with 'Measurement period for each parameter at each site…'.
** we will correct this.

228: Consider 'data' instead of 'amount'.
** we will correct this.

241: Consider 'hydrologic' instead of 'response'.
** we will change this.

243-4: Consider replacing 'Variability was also more sensitive under forested land cover…' with 'There was higher variability in volumetric soil moisture under forested land cover…'.
** we will change this.

Figure 4: It looks like the winter 2020 peak groundwater level at both sites is slightly higher than 2019. Could this suggest recovery?
** yes but in a tiny way. In 2022 groundwater levels were down already again. However, we will add and mention this slight recovery in 2020.

Figure 4: I'm not sure if this really matters, but the panels are introduced out of order in the caption.
** we will correct this.

Figure 5: The panels could be stretched out a bit more vertically. This would especially help with seeing some of the detail in panel (c). The titles above each panel could be repositioned beside the a), b), c), etc. to achieve this.
** we will correct this.

312: There was not a lot of information given in the methods on data quality control. I recommend including the accuracy of all the measurements in the dataset.
** we will include this into the data file.

[Figure]

**Figure 1: The Demnitzer Mill Creek catchment and its location within Europe and Germany. Measurement types are indicated in the legend, with red indicating no isotope measurements, black and purple indicating isotope measurements, and purple additionally indicating sap flow and sap isotope measurements. Meteorological measurements at Neu Madlitz were conducted by the German Weather service (DWD Deutscher Wetterdienst).)**

---

## Author Comment (AC2)

Tetzlaff et al. have posted a strong dataset with hydrologic and isotopic data from several compartments representing blue water and green water from a mixed land use catchment in northeast Germany. The dataset is strong and useful – I could not benefit fully due to problems I had accessing the metadata.
** thank you for the positive comments. Many apologies though that you couldn't access the file (more detailed response below). However, we double checked and we can download the metadata 622 und 623 (so hopefully this was just a temporary problem or unfortunately related to the PC of the reviewer?). If required, and the reviewer still has issues, we would appreciate it if one could report the error message to us and/or to the support for the FRED repository at fred@igb-berlin.de .

While the data quality appeared to be strong they could be better organized and made to be more intuitive to follow with better column labels. Even if explanations are present in the metadata, more intuitive headings could avoid making the search. Reviewer 1 did a thorough job of noting these instances, so some of my comments may overlap. I've divided my comments onto editorial comments on the text followed by comments pertaining to the data.
 ** yes, we will substantially revise and correct the data file (indeed, we have done this already and will upload the final, revised data file version once all reviewer comments, and potential comments by the editor are addressed and incorporated).

Editorial suggestions:
General: This is described as a 2-year dataset but there were references to groundwater levels and stream discharge from 20 and even 40 years ago. The text made no mention that the older data would be posted, but they are there on the website. It would be good to mention up front that you focused on the richer, newer data, but these "historical perspective" data are also posted, and particularly with the unusual drought during these two years, the older hydrologic data provide nice context.

Line 22. Suggest to delete "though."
** will be corrected.

Line 23. Parallel construction: "we" should consistently be within or outside of each numbered clause
** will be corrected.

Line 82. Drop "s" from "lowlands."
** will be corrected.

Line 92. Fix "data of two years of data…."
** will be corrected.

Line 155. Suggest either 'Measurements were made' or 'Throughfall was collected.'
** will be corrected to "Throughfall was collected…"

Line 156. 10-m
** will be corrected.

Line 160. "18 sensors"

** will be corrected.

Line 170-171. Suggest "Water level measurements were established…."
** will be corrected as suggested

Line 210. Spell check: "being"
** will be corrected (thanks for spotting!)

Line 226. Delete comma after "including"
** will be corrected

Line 236. Change "is" to "in"
** will be corrected

Fig. 2b. Start of record, July 2018 – are these good data? Throughfall >>rainfall only for this short period?
** Throughfall was collected weekly from the throughfall samplers starting in July 2018. Throughfall sampling continued weekly into the summer of 2020. Respectfully: the authors are not clear why the reviewer believes that this data is suspect as there were five (5) throughfall samplers for comparison throughout the Forest A site.

Line 243-244. more sensitive = greater?
** will be corrected (as also commented by reviewer 1).

Line 253. Consider simplifying sentence by deleting "despite" and "this"
** great suggestion - will be corrected

Line 278. Variation damped? (not samples)
** will be corrected to "though in deeper soil, the isotopic signal was more damped"

Line 301. data are
** will be corrected

Line 310. Spell check: "complementary"
** will be corrected

Data:
Like Reviewer 1, I had some confusion over the data versions. My comments were originally on the file in 623, but then I saw the author response to Reviewer 1 that site 622 had the revised dataset. The two versions looked the same to me, but if my comments are off it could be a version confusion issue.
** this was related to the doi. However, by now we have corrected the file and will upload the new file as soon as all reviewer and editor comments are addressed.

Unlike Reviewer 1 I could not download any metadata. Both the .eml and.xml file downloads were blocked.
** We double checked and we can download the metadata 622 und 623 (so hopefully this was just a temporary problem or unfortunately related to the PC of the reviewer?). If required, and the reviewer still has issues, we would appreciate it if one could report the error message to us and/or to the support for the FRED repository at fred@igb-berlin.de .

The General "readme" page at Site 622 says all sample times are CET, but makes no mention of accommodations (or not) for the summer time change. This may be clarified elsewhere but should be stated here.
** The time in all recorded date-time reflects the local daylight saving time corrections. This has been updated in the readme of the datasheet.

In the Hasenfelde Met_data tab, each parameter has an average, max, and min value for each hour (nice!), but the wind speed and direction are not specified. Are these averages or instantaneous values on the hour?
** The wind speed and direction are aggregated as data averaged over the hour. A note has been added to the readme in the data sheet for clarification.

The precipitation isotope tab states (in readme text in upper left) that aside from the daily Hasenfelde site, other sites might be event or weekly – is there any indication of which samples are weekly? This applies to Throughfall as well, its tab states throughfall was collected on the same schedule as precip.
** As per comments from Review 1, this has been clarified in the readme sheet in the datasheet.

On the Discharge tab, for Denmitz Mill, using the rating curve in cell F5, I can't reproduce the flows from the water levels.
** This has been clarified in the readme sheet. The rating curve shown in the sheet was established in 2007 after the relocation of the pressure gauge. The relocation was slightly downstream of the original location mandating a reestablishment of the rating curve. The original rating curve (2001-2007) has been added to the readme sheet.

In the Discharge_Isotope tab, the readme upper left says sample type is specified in site name but it is actually in the Type: field in the header for each site. And in this field (Row2), Grab" in grab sample is misspelled.
** This has been corrected (typo and description).

Groundwater tab: "Div/0 error in cells for GW4, March 2019. I don't understand readme note in upper left corner about only showing data for 2018, as all sites go to 2020. Duplicate identical data columns could be removed.
** The readme description has been corrected and all missing data (Div/0) has been corrected with NaN.

Forest A and Grass A SMC tabs – These tabs are hard to decipher – headers are minimal, sites are overly abbreviated where there is ample space to expand, and there are multiple columns with identical labels. Even if these are described in metadata elsewhere there should be more information directly in the tab to help decipher what is being presented. Is there a difference between Soil Moisture and Average Soil Moisture?
** These tabs have been updated for clarification. Headers have been improved for clarity, with an additional description of headers in the readme file. This includes the description of average and non-averaged soil moisture.

Forest A and Grass A Isotope tabs – Reporting of units in column headings is uneven, for instance columns J and K, grav_soil_moist (no units given) and water content g (just g? Not per unit volume?)

** the units have been updated with some further explanation added to the readme sheet.

Vegetation_isotopes tab, row 21 has data beyond where headers stop.
** Corrected

---

## Author Response (AR2)

**Public justification (visible to the public if the article is accepted and published)**:
The manuscript and dataset present a very comprehensive case for ecohydrologic studies. Some revisions are required as stated to the authors directly.

Additional private note (visible to authors and reviewers only):
Dear Dörthe and co-workers,
Please accept my apologies for the late response delayed over the holiday season and various deadlines on my side. Thank you again for submitting your work to ESSD. Since the reviewers indicated that only minor revisions were required, it is on my side to review the revised version.

**Dear Conrad,
Please find our detailed responses below. Unfortunately, some of the comments resulted from the fact that we incorporated all comments by the reviewers (which were partly contradictory now). Hopefully we addressed all to your satisfaction now and that the paper can be accepted for publication.
Regards,
Doerthe Tetzlaff on behalf of all coauthors.

DATASET:
The DOI 10.18728/igb-fred-623.0 given in the manuscript, does not link to the actual file, which appears to reside in https://fred.igb-berlin.de/data/package/622 (also stating the wrong DOI there). Please revise and correct the data storage, which is required to guarantee longterm workable data access. Alternatives to you FRED in-house data center are e.g. https://www.pangaea.de/ or https://dataservices.gfz-potsdam.de/portal/index.html .
** as written in our original response file: once all final edits are done in the data file (incl yours below) we will now update and finalise the doi. The final link to the correct data package is now https://doi.org/10.18728/igb-fred-813.1" along with the associated change(s) in the manuscript and any other metadata.

Data description: Despite all the value of your data, it is still not very easy to reproduce how your data is structured and how to actually use it. You now include 19 lines readme, which really requires very intense study of the data and your manuscript to decipher. The included citations of Dubbert et al. (2013, 2014) are only given in the ESSD manuscript. In your manuscript you cite the dataset (see below L154ff) and in the dataset you cite this manuscript for details. However, the details are rather sparse in both…
** This is unfortunate that the editor gets that impression as this was the result now of the comments we implemented following the reviewer's comments. These extra readme lines have been removed from the dataset again now, referring the readers to the manuscript for more details on the measurement methods. We actually added those during the revision requested by one of the reviewers in the last round. But we can see how this would cause confusion so it makes more sense to remove it. We removed the citation in the data base BUT of course refer to it in the manuscript.

What do you mean with "Times all *reflect* local daylight saving time corrections."? So it is CET with pr without daylight saving?
** This has been corrected by clarifying that DST times for winter are equivalent to UTC+02:00 and summer are UST+03:00

How were the CRDS measurements calibrated and corrected for drift? How many repetitions and simple mean or any hyperbolic estimates?
** we added that information in the manuscript text (referring the reader / user from the datasheet to the manuscript)

The sap flow data appear hard to reproduce. You have installed 2-4 TDP probes in 12 trees of different species (of which 4 are named). An unspecified subset of this data has been averaged and is given here as L/day. As you know, there are quite some processing steps to derive a volumetric flux from the measured temperature differences including an estimate about the radial distribution of the xylem flow velocity field, xylem cross section, etc.

** yes, of course. However, we felt such extensive method description can't give so much method data. Now, we extended this in the manuscript and also refer to Kleine et al., where the method is described in more detail. We also added some additional description in the readme file as well (all trees were considered). The additional information also clarifies that the data presented is for a forest stand not for an individual tree (we've modified the sheet name to clarify this also). We have only ever used the forest stand transpiration, not transpiration for individual trees so we don't think it's necessary to include any further data here.

I have no idea what to do with this information: "Groundwater (tab name) data only begin in 2018, corrected by Jonas F. These are separate from the long-term data as the levels are reset in 2018". Why are these records separated?

**These datasets are separated as they were re-established in 2018, with a change in data collection and data management. To avoid confusing these datasets, they have been separated. This description has been added to the readme file.

What means forest north/south?

** this was an ID terminology. We have clarified this in the readme file of the dataset. It is included to distinguish the datasets (this refers to fn of fs as id labels).

Positions: UTM33 is not a unique coordinate system. Please report the ESPG code. Is it EPSG:25833? (https://epsg.io/25833)

**Corrected

Would it be an option to either clarify the meaning of the given column names or simply avoid abbreviations? Maybe a brief description above each table would also help clarification?

**An additional sheet has been added for further description of the abbreviations of the headers of other data sheets. For us – and respectfully - avoiding abbreviations is not an option, the headers would be too long. We were advised to remove any descriptions from the datasheets in the previous revision (hence the longer ReadMe) so this is not an option either.

MANUSCRIPT:
Abstract and Dataset: Pls. check our submission guidelines. https://www.earth-system-science-data.net/submission.html It is required to include "a functional data set DOI and its in-text citation" in the abstract.

** added

L119: There is an extra closed bracket. Pls. remove.

** corrected

L154ff: I do not quite understand why you cite the doi-url seven times. I suspect that you intended to refer to the different sheets in the excel file?

** because the editor asked for a repeated citation in his original comments to us prior external review. Statement by the editor back then: "Technically, your data has a DOI (https://doi.org/10.18728/igb-fred-623.0) and should be cited through this in the manuscript. You will be asked for this in the copy editing process anyway." We THEREFORE added the citation of the

doi throughout the text. And no, its just the one doi. We respectfully decided to leave this as it is now.

L179ff: Why are "Micro 10m and Baro" and "Van Essen Instruments" in two separate brackets? I suspect that you refer to the respective autologging DIVERS? But why do you then refer to the ATP10 AquiLite for the atmospheric pressure correction, which I suspect is simply done in the DIVER software? I find the structure of this paragraph difficult to comprehend.
** we apologise for this confusion This was wrong and is now corrected.

L189: Groundwater is a new paragraph?
** corrected.

L195ff: You have referred to isotope sampling in streams and at the AWS earlier. Maybe it is easier to understand when this is all given here? This Subsection could also include more details about the actual CRDS analysis (repeated sampling, standards, drift correction, …).
** yes good suggestion. We moved the text and also added more information on the CRDS analysis.

Fig 3: Really difficult to see. Can this become a little larger avoiding the large margins? I do not understand the 2nd part of the caption. What do the boxes in b show? Weekly precip somewhere minus throughfall? Each box consists of 5 data points? At this scale, maybe monthly aggregates would be more legible? Maybe then monthly precip would also be a better reference than the cumulative curves in which I cannot discern throughfall from DM and in which I have no idea what the green lines are?
** we are wondering if you are they are mis-interpreting this. We clarified in the caption now that "Precipitation at Bruchmill (nearby) was used as to calculate weekly interception loss." Re: overlapping lines, we don't think it's an issue that they overlap, but what's there to see if they overlap anyway, it's showing that they are consistent. Respectfully, aggregation also doesn't make much sense as we want to show the variability rather than a smoothed signal as it showcases the amount of data collection.

Fig 4: This figure is quite complex but gives a nice overview about your data. The quality is rather poor which I hope is only a conversion issue. Would it be possible to merge Fig 3 and 4 into one like Fig 2 and 5 (with panes using the full width of the page)?
** as the editor points out: Fig 4 is already very complex. Also contains different data than fig 3 so merging them would be very difficult. We respectfully ask that we can keep them separate (and yes of course the figures are in the original format much higher and better resolution).

L356ff: This appears to include copied line numbers from earlier versions?
** corrected

---

## Author Response (AR3)

**Public justification (visible to the public if the article is accepted and published)**:
The manuscript and dataset present a very comprehensive case for ecohydrologic studies. Some revisions are required as stated to the authors directly.

Additional private note (visible to authors and reviewers only):
Dear Dörthe and co-workers,

I am sorry to keep asking for some revisions. Please be assured, that I really do not take it lightly to guide the process of publishing your data.

Dear Conrad,
Please find below and attached our 4[th] round of revision. All revisions were "with minor comments". Again, we tried to response to and address all comments carefully. Some of these comments were new and not raised in the 3 rounds before.

In addition we incorporated all the comments and suggestions received during our phonecall and subsequent email from the EiC. In summary:
We have heavily revised the data sheet and the meta data file.
We revised Tale 2 in the manuscript.
We revised Figures 2, 3 and 4. We revised the captions for Figures 5 and 6.

We also incorporated all final comments by the EiC and the final comments (via email) by yourself.
We would like to thank the EiC and yourself for your time and efforts with our manuscript,

kind reagrds,
Doerthe Tetzlaff on behalf of all coauthors

I have discussed with other editors in ESSD and they too find it worthwhile to not simply waive the manuscript.
** this might be a "translation" issue but we are stunned that the Editor suggests that this paper would be "waived" in any way considering this is the 4[th] round of revision!

It appears that we have to become clearer in our submission guidelines, what we expect a data publication to be like and how they fill the niche between the documented dataset publication on a repository and the scientific papers arising from the data.
** we agree.

For the repository, I simply wanted to make sure that you double check our requirements https://www.earth-system-science-data.net/policies/repository_criteria.html and the different links in both, the manuscript and your repository pointing to the old version are updated.
** we have done this.

It appears that I caused some confusion about the data description.
** we agree.

The misunderstanding might be, that the dataset itself requires a technical description which enables other researchers to technically understand what is to be found in the dataset and what is behind each of the data values. In that terms the "readme lines" have been much appreciated and went into the right direction and should be extended to really clarify all the tables. Complementary,

the ESSD manuscript can remain at the current level of detail summarising the scientific value of this dataset. And it makes it much easier to link to specific parts in your dataset in L154ff.

**we have done this. As with the editor's other comments, the authors have further elaborated on the header names and descriptions which additionally fulfil this request.

(I still neither technically nor aesthetically understand why the repeated citation of the whole dataset is chosen).

** corrected as this was based on a misunderstanding in the first round of reviews.

Since your data repository did not curate your submission, it is unfortunately at my side now to ask you for a technical description of your data. Basically the readme can include and extend the 19 lines suggested earlier plus it should clarify the technical content of each sheet and each column. This includes to decipher the chosen abbreviations etc. and is a basic requirement of the FAIR principles.

** addressed.

Fig. 3: Thank you for the clarification. Again, I might have sparked confusion: My issue in the first place is that the plot comes in rather poor quality so that I cannot see some of the details. From what I can read, there are more line colours than in the legend. I suspect that the green lines are the throughfall collectors in Forest A? Is precipitation Berkenbruck actually throughfall precip (see classification in Fig 2)? I am ok with weekly box plots, but I suspect that the plot will be much easier to read when the large whitespaces left and right are dropped. Moreover, I respect your decision to leave Fig 3 and 4 as separate plots – although I still think that the arrangement as in Fig 5 would be easier to read.

** we revised fig 3 and fig 4. The caption in fig 5 was clarified.

Tables: I could not find any link to the tables in your manuscript. While table 1 clearly can be linked in L128, I am not sure if table 2 is really required as substantial part of your manuscript. Since most of its content is depicted in Fig 2 it could become an appendix table? Or maybe simply a file/sheet in your dataset?

** included now (but again it would have been helpful to receive this comment / request in round 3). As in none of the previous rounds neither the editor nor the reviewers wanted us to remove table 2 we respectfully decided to leave that table in.

I hope that I could resolve some of the confusion which occurred meanwhile and I hope that you find my suggestions helpful and easy to incorporate. I am looking forward to the publication of your paper.

** You probably can not imagine how much we would look forward to ever seen this paper published.